



# W.A.T.E.R. – a structured approach for training on advanced measurement and experimental research

Margaret Chen[1], Rui Aleixo[1,2], Massimo Guerrero[3], Rui Ferreira[2,4]

[1]Hydrology and Hydraulic Engineering, Vrije Universiteit Brussel, Brussels, 1050, Belgium
[2]CERIS-Civil Engineering Research for Innovation and Sustainability, Lisbon, 1049-001, Portugal
[3]Department of Civil, Chemical, Environmental, and Materials Engineering, University of Bologna, Bologna, 40136, Italia
[4]Instituto Superior Técnico, Universidadede Lisboa, Lisboa, 1049-001, Portugal

*Correspondence to*: Margaret Chen (margaret.chen@vub.be)

**Abstract.** W.A.T.E.R. stands for Workshop on Advanced measurement Techniques and Experimental Research. It is an initiative started in 2016, in the scope of the Experimental Methods and Instrumentation (EMI) committee of the International Association for Hydroenvironment Research (IAHR) aimed to advance the use of experimental techniques in hydraulics and fluid mechanics research. It provides a structured approach for the learning and training platform to postgraduate students, young researchers, and professionals with an experimental background in fluid mechanics. It offers an
opportunity to learn about state-of-the-art instrumentation and measurement techniques and the latest developments in the field by partnering with manufacturers. The W.A.T.E.R. brings together academics, instrumentation manufacturers and public sectors in a structured setting to share knowledge and to learn from good practices. It is about training people that already have certain knowledge and skill level but need to go deeper and/or wider in the field of measurement and experimental research.

## 1 Inception of W.A.T.E.R.

There were several motives underpinning to create an event dedicated to training of researchers and professionals in the field of instrumentation in hydraulics and fluid mechanics. Among the hydraulics research community, it is identified that there are needs: i) to provide continuing education and training in advanced and state-of-the-art measurement techniques ii) to offer training in experimental research topics that usually are not taught in hydraulics and fluid mechanics curricula; iii) to
strengthen the collaboration with industries developing new experimental techniques and data analysis routines in the field of hydro-environment studies; and iv) to promote an international environment where researchers coming from different backgrounds can learn and share good practices in measurement techniques and experimental research. Since items iii) and iv) fit within the scope of the Experimental Methods and Instrumentation Committee (EMI) (https://www.iahr.org/index/committe/1), its leadership team (2015-2017) fully supported the idea of Margaret Chen to
organize such event. The first edition of the W.A.T.E.R. took place in 2016, in Oostende, Belgium, organized by the Vrije





Universiteit Brussels. Since 2016, 5 editions were organized; due to the pandemics the edition of 2020 was postponed to 2021 and took place in Bolzano, Italy (W.A.T.E.R. https://water2020.events.unibz.it/). Currently, the 2022 edition is under preparation and will be organized by the Instituto Superior Técnico of the Universidade de Lisboa. Table 1 presents the W.A.T.E.R. events already organized and the upcoming one. The W.A.T.E.R. is fully aligned with the strategic plans of both

the EMI and of the International Association for Hydro-Environment Engineering and Research (IAHR) (https://www.iahr.org/index/committe/1). Hence, the current leadership team of the EMI committee has in its agenda to include W.A.T.E.R. as a regular event applying the W.A.T.E.R. standard procedures and guidelines for selection of future host- organizations and venues.

**Table 1: W.A.T.E.R. editions, the locations and number of participants.**

| W.A.T.E.R. editions | Organizer | Location | Number of participants |
|---|---|---|---|
| 2016 | Vrije Universiteit Brussels, Belgium | Oostende, Belgium | 12 |
| 2017 | Vrije Universiteit Brussels, Belgium | Oostende, Belgium | 14 |
| 2018 | Vrije Universiteit Brussels, Belgium | Oostende, Belgium | 14 |
| 2019 | University of Bologna, Italy | Bologna, Italy | 19 |
| 2020 | Postponed to 2021 due to covid | | |
| 2021 | University of Bolzano, Italy | Bolzano, Italy | 14 |
| 2022 | IST - Universidade de Lisboa | Lisbon, Portugal | |

## 2 The ethos of W.A.T.E.R.

The fundamental characteristic of the W.A.T.E.R. is its hands-on philosophy. From the start it was decided that the course would have a strong practical component with laboratory and field measurement sessions. In the W.A.T.E.R., besides the theory associated to each technique/instrument, the participants are encouraged to practice the different techniques in both

laboratory and field conditions. To achieve this goal, different assignments are proposed, allowing the participants to measure real flows with different techniques, and to process and analyze the measured data, while at the same time working in groups of diverse scientific and cultural backgrounds. Another characteristic of the W.A.T.E.R. is to introduce participants not only to established techniques, such as 2D Particle Image Velocimetry, but also to present the most recent developments and latest technologies. Different manufacturers, such as ILA5150 (ILA5150 www.ila5150.de) and UBERTONE

(UBERTONE www.ubertone.com) partnered with W.A.T.E.R. and present their cutting-edge know-how to the W.A.T.E.R. An example was the introduction by UBERTONE, in the 4[th] edition of the W.A.T.E.R., of its novel instrument (launched in 2019) the Acoustic Doppler Velocity Profiler (ADVP) allowing for the simultaneous measurement of the profile of two velocity components. At the same edition, ILA5150 presented its most recent version of PIV software, as well as an





illumination setup based on high energy LEDs thus avoiding the use of LASERs and making their PIV system more portable.

In addition, since the W.AT.E.R. is being co-organized with universities, it allows to be recognized for learning credits (e.g., the European Credit Transfer and Accumulation System - ECTS) and thus provides support to its postgraduate participants completing part of their advanced training leading to doctoral degrees.

Finally, another feature of the W.A.T.E.R. is providing advanced training at reasonable costs. The fees are meant to cover the essential material costs of each edition.

## 2.1 W.A.T.E.R. Organization

Figure 1 depicts the W.A.T.E.R. typical schedule. The first day and half are filled with the theoretical part of each technique. Then it follows another day and half that are dedicated to the parallel laboratory sessions, keynotes and masterclass. Masterclass is the time allocated to participants who, on a voluntary basis, wish to present their research and receive feedback from the lecturers and other participants. A full day is dedicated to field measurements and the last day is dedicated to preparing the reporting (morning) and evaluation (afternoon).

Versatility is one of the main features of the W.A.T.E.R. and alternative plans are ready in case the original plan has to be modified due to, for example, the impossibility of field measurements because of unforeseen bad weather. Time is also allocated to social events, such as the ice break gathering in the evening of the first day, and the W.A.T.E.R. diner. Furthermore, the interactive coffee-breaks and lunches are made to allow participants to discuss and interact.

## 2.2 W.A.T.E.R. Outreach

W.A.T.E.R. has an official website (W.A.T.E.R. website) to provide the relevant information about the ongoing activities, deadlines, application procedures, etc. To reach out to its potential members, the W.A.T.E.R. has also a presence in Facebook (W.A.T.E.R. Facebook) this allows an easier communication and interaction with potential participants.

## 3 Measurement techniques and experimental setups

In hydraulics and fluid mechanics several measurement techniques exist and are frequently used. At the W.A.T.E.R. Summer School emphasis is given to acoustic-based and optical-based techniques. The former deals with velocity measurements based on the acoustic Doppler effect and the latter with velocity measurements based on Laser Doppler and imaging techniques such as Particle Image Velocimetry (PIV) and Particle Tracking Velocimetry (PTV). This choice stems from the fact that these families of techniques are almost ubiquitous in many laboratories and field monitoring stations.

Acoustic-based instruments allow measuring the fluid velocity in a point or in a profile along the instrument axis (Muste et al. 2017). It is possible to achieve sampling frequencies of 100 Hz, allowing these instruments to measure turbulent





fluctuations. Acoustic instruments are usually sturdy and well built, allowing them to be deployed in the field and work under rough flow conditions and with turbid and poach fluids.

Optical based techniques, in particular PIV (Raffel et al, 2010, Adrian and Westerweel, 2011), rely on image acquisition and processing. PIV allows measuring the whole flow in the image plane and thus constitutes a valuable tool for flow measurements. Its most common use is in the laboratory as the need for optical access and delicate optical alignments hinders the frequent use in field conditions, although it is possible, under special circumstances, deploy it also to the field. For field applications PIV algorithms are often applied to surface images to measure the surface velocity using the same

algorithms (Jodeau et al. 2017, Lewis and Rhoads, 2018).

Particle Tracking Velocimetry (Hassan and Canaan, 1991, Capart et al. 2002), from a point of view of image processing, is also presented during the W.A.T.E.R. The PTV allows the participants to complement the training of PIV with a similar, but different technique.

In addition, since 2019, Laser Doppler Velocimetry (LDV) is also introduced as an example of a point-wise technique. LDV

is often used, over other forms of flow measurement, as a standard for reference and calibration, given that it constitutes the greater accuracy to measure the fluid velocity at a given point in turbulence studies and generic fluid mechanics measurements.

By partnering with two key players of acoustic and optical solutions, UBERTONE and ILA5150, W.A.T.E.R. provides its participants access to the state-of-the-art techniques.

**3.1 Experimental setups**

The experimental part of the W.A.T.E.R. Summer School is typically carried out in the hydraulics laboratory of the organizing university. This allows the participants to use already existing installations as well as provides examples of didactical and research relevance. Experiments are performed in setups that allow the participants to be aware of the potentialities and limitations of each technique. Some setups provide classical problems such as the flow around a cylinder

(Figure 2), including the calculation of the Strouhal frequency of vortex shedding and mean drag and lift coefficients. In this setup PIV is often used, allowing measuring the planar flow field around the cylinder for different Reynolds numbers. Participants start the respective assignment by setting up the parameters of the PIV, namely the time between consecutive frames, focus of the image, calibration, etc; then they proceed to acquire flow images and finally they process the data to obtain the velocity field and by post-processing the measured velocity field, they extract the generated vortices frequency to

calculate the Strouhal number and compare it with the literature. Another setup is aimed at boundary-layer turbulence (Schlichting and Gersten, 2017), allowing the participants to determine the mean velocity and second-order moments using, for example, an ultra-sonic velocity profiler (Figure 3).

It is common to compare two different techniques. For example, at the 4[th] Edition of W.A.T.E.R. in Bologna, the novel ADVP was used to measure a boundary layer flow, at the same section as the LDV, allowing participants to conduct a direct

comparison and to discuss advantages and disadvantages of each technique (Figure 4).

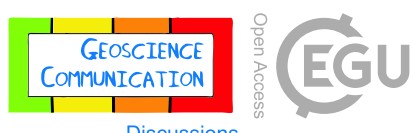

During the experimental courses, the lecturers are present and guide each session on the technique they lectured, allowing a better coupling between theory and practice. The discussion is further enhanced by the presence of personnel from the industries that provided the instruments.

To optimize the time and laboratory spaces, different setups run in parallel, this allows the maximized experience of each
participant to have hands-on training of each technique lectured in the W.A.T.E.R. program.

## 3.2 Field measurements

Field measurements are also part of the curriculum of the W.A.T.E.R. Summer School. Field measurements in Oostende included measurements in the Northern Sea in Research Vessel Simon Stevin of the Flemish Marine Institute (Research Vessel Simon Stevin) (Figure 5). Field measurements in Bologna were made possible thanks to the Po River agency (Po
River Agency) (Figure 6). Field measurements allow participants to be in contact with different users of experimental techniques, specifically state agencies. It allows them to understand the limitations and challenges of field measurements and to get in touch with novel approaches, such as the use of unmanned vehicles (Figure 7).

By partnering with local public organizations, the W.A.T.E.R. provides access to state-of-the-art research facilities and instrumentssuch as the unmanned survey vehicles (USV), and the Proambiente's USV (Figure 7).

## 4 Evaluation

The participants are evaluated at the end of each event before issuing the respective certificate with the ECTS. Evaluation is made of three parts: i) questionnaire about the different techniques; ii) oral presentation of the assignments and iii) questioning session (Q&A)  after the presentation. Although participants can prepare their assignment presentation during the W.A.T.E.R., the morning period of the last day is reserved for report preparation. Debate between lecturers and
participants is encouraged in this period, allowing the participants to clarify relevant questions.

In order to improve from edition to edition, at the end of each edition, an evaluation form is distributed to the participants who can anonymously express their critiques and remarks about the event. Figure 8 depicts the results of the evaluation made by the participants.

## 5. Profile of the participants

The profile of the typical W.A.T.E.R. participant has a mean age of 29 years old and pursuing a PhD. However, about 25% of the participants are professionals who need to upgrade or improve their competences in the experimental hydraulics and measurement techniques domain. Among the participants about 65% is male and 35%  is female. 75% of the participants come from Europe, 12% from America, and13% from Asia. Figure 9 depicts the cumulative results of wherethe participants come from.





## 6. Conclusions

W.A.T.E.R. is a training event designed to offer training in advanced measurement techniques in hydraulics and fluid mechanics. The organization of the event is made in cooperation between universities and instrument manufacturers and the advantages are twofold: i) the recognition of the training in terms of ECTS; ii) the access to state-of the-art techniques. Furthermore, it provides a welcoming and friendly environment where participants from different backgrounds can interact and improve their skills. Social events are also part of the program providing time to networking and relaxing.

The light organization of the W.A.T.E.R., and a program that has been tested and improved for 5 consecutive editions allows it to easily adapt to different scenarios and circumstances and to provide meaningful and useful contents to the participants. The current leadership team of EMI committee is determined to support W.A.T.E.R. as a structuring event of the committee and a regular occurrence in the IAHR agenda.

## Author contributions

Conceptualization, M.C. and R.A.; writing – original draft preparation, M.C. and R.A.; writing – review and editing, M.G. and R.F.; all authors have participated in the finalization of the manuscript.

## Competing interests

The authors declare that they have no conflict of interest.

## Acknowledgments

Acknowledgements to Marie Burckbuchker and Stephan Fischer from UBERTONE, and Frank Michaux from ILA5150 for the technical support given to the W.A.T.E.R.

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





|  | Monday | Tuesday | Wednesday | Thursday | Friday |
|---|---|---|---|---|---|
| 8:30-9:00 | Welcome | Innovative experimental techniques in fluid dynamics; | Seminar 2 | Field Trip<br><br>Advanced field measurement techniques | Report preparation and lab sessions assignments |
| 9:0 0-9:30 | PIV |  |  |  |  |
| 9:30-10:00 |  |  |  |  |  |
| 10:00-10:30 |  |  |  |  |  |
| 10:30-11:00 | Coffee-Break | Coffee-Break | Coffee-Break |  | Coffee-Break |
| 11:00-11:30 | PIV | Acoustic Velocimetry | Seminar 1 |  | Report preparation and lab sessions assignments |
| 11:30-12:00 |  |  |  |  |  |
| 12:00-12:30 |  |  |  |  |  |
| 12:30-13:00 |  |  |  |  |  |
| 13:00-13:30 | Lunch | Lunch | Lunch |  | Lunch |
| 13:30-14:00 |  |  |  |  |  |
| 14:00-14:30 | PTV | Seminar | Parallel Lab Sessions 1/2/3/4/5 |  | Evaluation for course diploma with 5 ECTS Interplay between W.A.T.E.R. team and local stakeholders |
| 14:30-15:00 |  | Parallel Lab Sessions 1/2/3/4/5 |  |  |  |
| 15:00-15:30 |  |  |  |  |  |
| 15:30-16:00 |  |  |  |  |  |
| 16:00-16:30 | Coffee-Break | Coffee-Break | Coffee-Break |  |  |
| 16:30-17:00 | Seminar |  | Seminar |  |  |
| 17:00-17:30 | LDA | Parallel Lab Sessions 1/2/3/4/5 | Master Class peer to peer discussion participants present their work and measurement issues (poster session) |  |  |
| 17:30-18:00 |  |  |  |  |  |
| 18:00-18:30 |  |  |  |  |  |

Ice Breaking Aperitif         Summer School Dinner

Figure 1: Typical schedule of the W.A.T.E.R. Summer School.

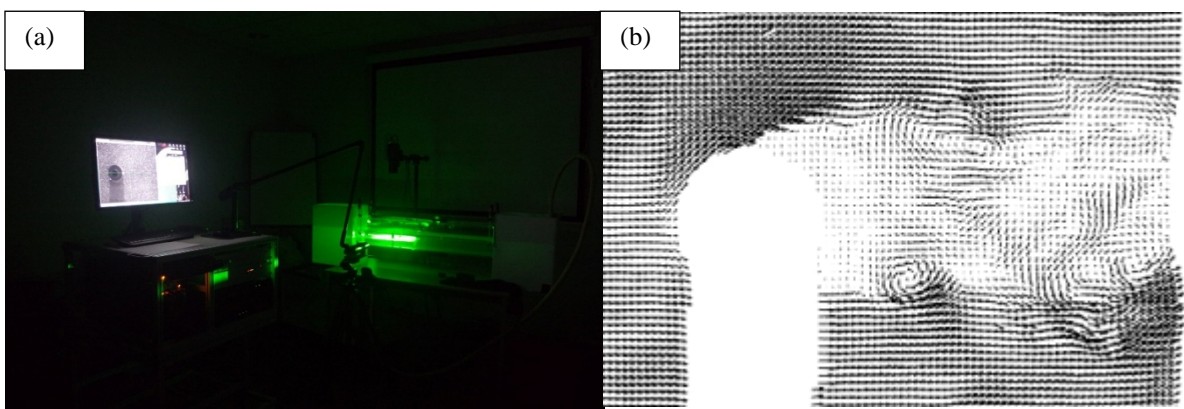


Figure 2: PIV measurements of the flow around a cylinder. (a) On the right the flume with a cylinder being lighted by the LASER and, on the left, on the monitor, the obtained flow image in real time. (b) The obtained velocity map. PIV system courtesy of ILA5150 (W.A.T.E.R. 2017).



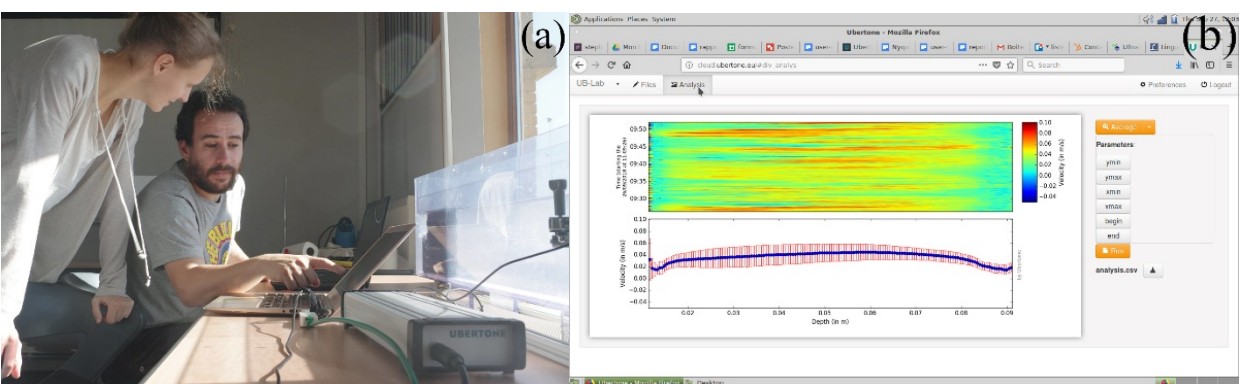

**Figure 3: (a) Hands on session measuring velocity profiles in a channel using UBERTONE's UB-Flow. (b) measured mean velocity profile (W.A.T.E.R. 2017).**

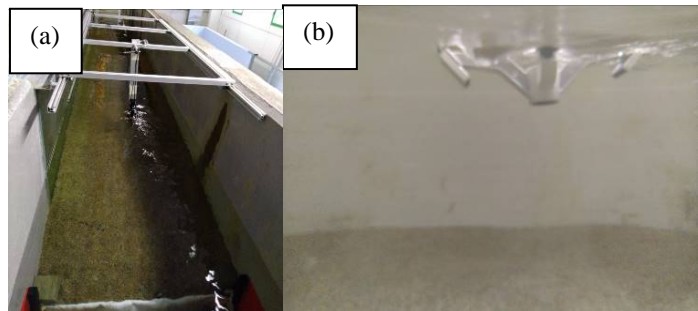

**Figure 4: Hands on session during the W.A.T.E.R. 2019 in the Hydraulics Laboratory of the University of Bologna (Italy): (a) and**
**(b) two-components velocity profile measurements with ADCP provided by UBERTONE.**

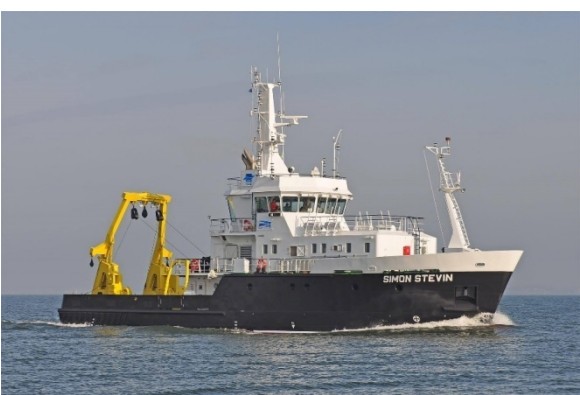

**Figure 5: Research Vessel Simon Stevin, courtesy of the Flemish Institute of the Sea (W.A.T.E.R. 2016 and W.A.T.E.R. 2017).**





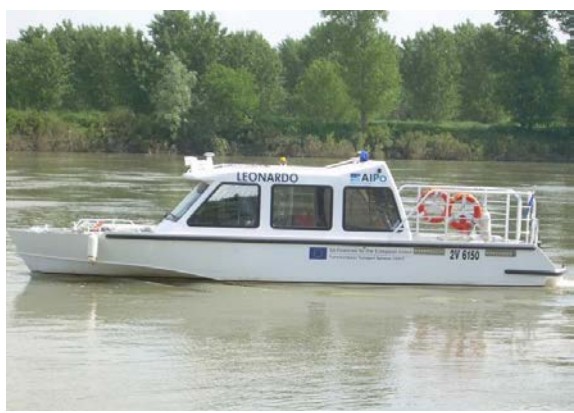


**Figure 6: Monitoring vessel Leonardo, courtesy of the River Po Agency (W.A.T.E.R. 2019).**

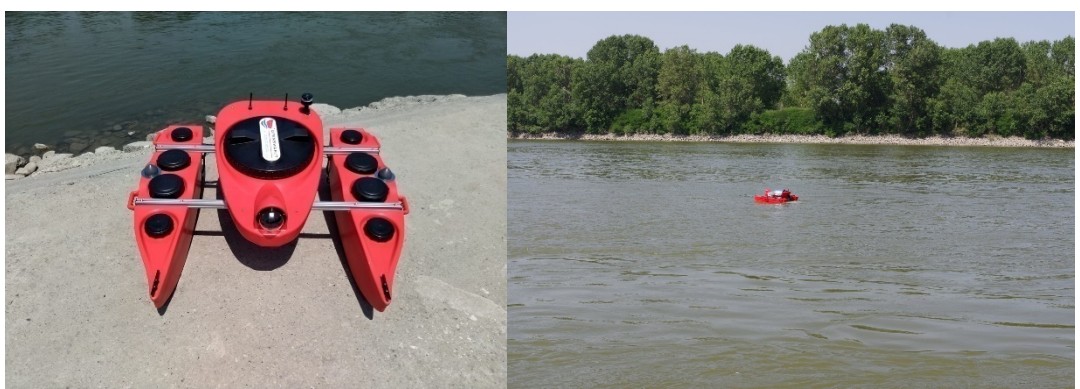

**Figure 7: Proambiente's Unmanned Survey Vehicle, during the field measurement session at Po River during W.A.T.E.R. 2019.**





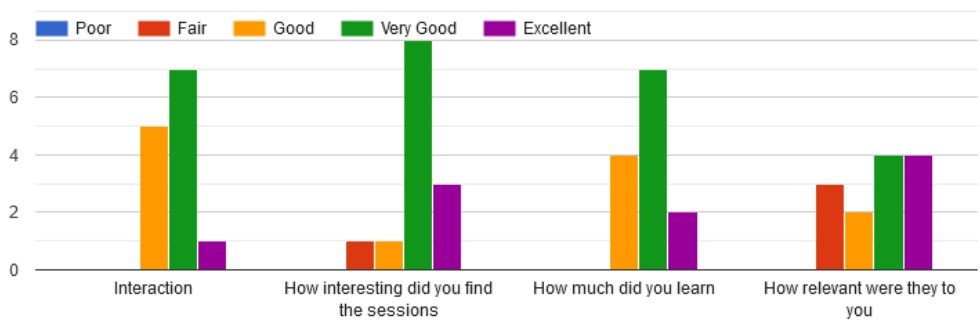

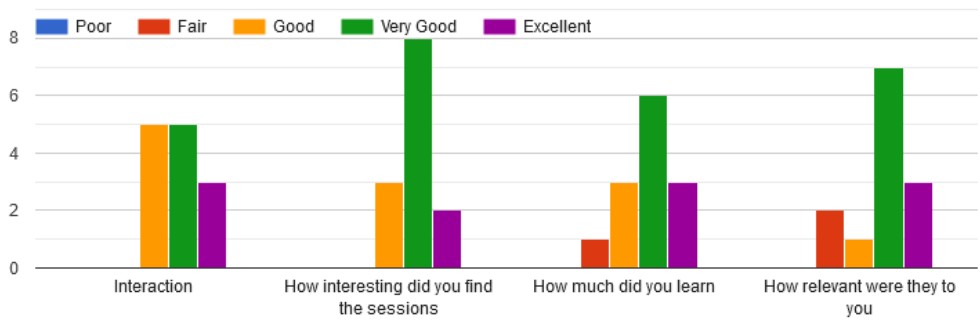


**Fig. 8.** Results of the evaluation by the participants of the W.A.T.E.R..

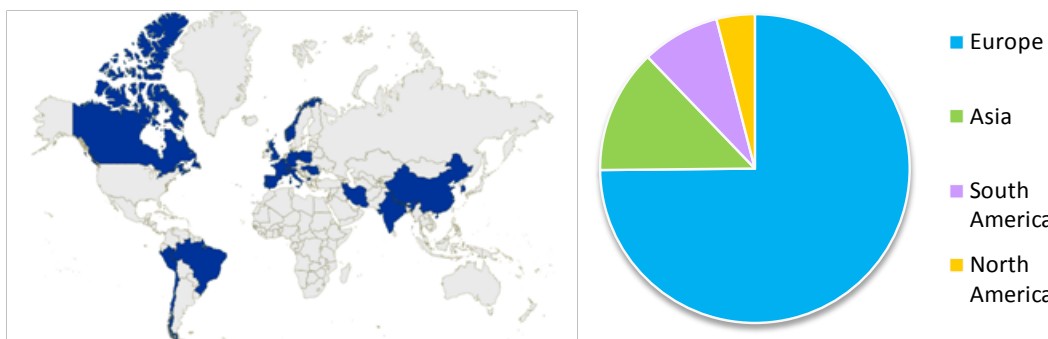

**Fig. 9.** The distribution of the W.A.T.E.R. participants (cumulative results).