# Peer review of "W.A.T.E.R. – a structured approach for training on advanced measurement and experimental research"

_Geoscience Communication, 2021_

## Author Response (AR1)

**Manuscript No.: gc-2021-47**
**Title**: W.A.T.E.R. - a structured approach for training on advanced measurement and experimental research
**Authors**: Margaret Chen et al.
**Special Issue**: Education and professional development in hydro-environmental engineering

**Date: Mar. 16, 2022**

Dear Editor,

Thank you very much for communicating the comments on our manuscript (MS) from the referees. We have revised the MS in accordance with the comments and suggestions from the referees. Here is our point-by-point response to all the general and specific comments.

**Referee #1**

- Your point (a): in the abstract you wrote line 13 about "the" learning and training platform, but in the paper W.A.T.E.R is described as "event" or as "Summer School" and not as platform. I recommend to classify this by only one type (e.g. workshop event and workshop series) and not using different terms

   Response:
   We agree and have revised accordingly using the W.A.T.E.R workshop series consistently throughout the MS.

- Your point (b): I propose to mention the main target group of PhD students in the abstract and introduction and not "late" on page 5 in chapter 5.

   Response:
   We have made the main target group especially doctoral students clear throughout the MS.

- Your point (c): in Table 1, the number of involved lecturer and of involved organizations might be added to underline the interinstitutional concept, the lecturers listed on the Webpage demonstrate this, but in the paper the information is missing. 2022 Portugal as country of the organizer is missing

   Response:
   The involved organizations are now mentioned in the MS before Table 1. A column with the number of lecturers has been added. Also, Portugal as country of the organizer in 2022 has been added in the Table 1.

- Your point (d): you mentioned ECTS but never wrote about the number of credits and related work load (e.h. in hours), maybe you can add at least for the last event in Bolzano the number of

ECTS achieved by the participants as example. How is the academic recognition done by the involved universities?

Response:
The academic recognition is realized in accordance with the education and other relevant regulations defined by the respective host university. Typically the study load is equivalent to 3-5 ECTS. 3ECTS were granted for the first three editions in Belgium, and 5ECTS were granted for the 4th (at Bologna) and the 5th (at Bolzano) editions in Italy. (Lines 62-67)

- Your point (e): Is teaching material prepared and distributed to the participants?

Response:
The teaching materials are prepared in advance and distributed to the participants during the event. (Lines 68-69)

- Your point (f): How can other institutions / IAHR members can benefit from this activity (I assume at least to send participants?)

Response:
Based on our recent experience, other institutions/IAHR members can benefit directly by, for instance:
- Sending their members for training.
- Participating by presenting state-of-the-art techniques (e.g. UBERTONE, ILA5150).
- Proposing a request to organize a W.A.T.E.R. event.

- Your point (g): line 58: what are reasonable costs? Maybe you give a fee example from the 2021 event including the covered costs such as accommodation, ....

Response:
Regarding costs, the W.A.T.E.R. initiative aims to be financially neutral. It is typically a 5-day event, and the fees are used to cover the major expenses including the travel costs of invited lecturers, catering, facilities, assurance, etc. The registration fees of the first 2 editions included the accommodation costs for the participants. Due to increasing costs of accommodation and diverse preferences from the participants, it was decided to drop this option and to exclude the accommodation costs from the registration fees from the third edition onward. The registration fee from the last edition was 490eur. This fee is quite competitive compared to the fees defined by other training events nowadays, for instance, an example can be found from (https://www.vki.ac.be/index.php/events-ls/events/eventdetail/522/-/on-site-lecture-series-introduction-to-ground-testing-facilities).

- Your point:  Fig 9 : numbers are missing ! what is shown: the nationality of the participation, or the countries of their actual university/institution ? Is a PhD student from India, doing his/her PhD at VUB counted for Belgium/Europe or as India ?

  Response:
  We have remade Figure 9 to make it clear. Fig 9 reflects the distribution of the W.A.T.E.R. participants according to their affiliations. In your example "If a PhD student from India, doing his/her PhD at VUB counted for Belgium/Europe or as India ?" In this example, this participant will be counted as a participant from Belgium and reflected on the figure in group Europe.

- Your point:  last link https://water2020.events.unibz.it/
  503 Service Temporarily Unavailable !!!

  Response:
  This link was deactivated and consequently it is removed from the reference list.

**Referee #2**

- Your comment:
  Please provide links wherever websites are mentioned (e.g. lines 73 and 75).

  Response:
  We have provided the links when websites are mentioned. Also, the links to the websites are all listed in the reference.

- Your comment:
  Line 80: referring to acousting and imaging velocity measurements the authors state:  "This choice stems from the fact that these families of techniques are almost ubiquitous in many laboratories and field monitoring stations." I am not sure if the statement is valid for experimental facilities on pressurized flows, maybe the challenges of applying such techniques in pressurized closed conduits could be mentioned.

  Response:
  W.A.T.E.R. benefits from the expertise and setups of the organizing institutions. There is an emphasis on open surface flows that stems from this fact so far. These families of techniques can both be applied to pressurized flows: imaging-based techniques will need optical access, but acoustic techniques such as UVP can measure through the wall, provided the acoustic impedance between the pipe and sensor is matched.

- Your comment:

Lines 85 to 97: acquisition frequency is mentioned for ADVP measurements but not for PIV, PTV or LDV, which is relevant and useful information. Would be nice if this could be included as well as accuracy thresholds.

Response:
Several measurement techniques were presented, and hands-on training of each technique was provided at the W.A.T.E.R. workshop series. The measurement frequency was achieved at 100 Hz for ADVP, 50 Hz for PIV, 100 Hz for PTV, 130 Hz (short time series) and 350 Hz (long time series) for LDV. This information is now included in the MS.

- Your comment:
Line 136: "from edition to edition..." Fig.8 does not provide the progress of participants' perception, trends could be depicted if possible.

Response:
The questionnaires have been evolved with the editions. We have been including new questions to assess the quality after each edition. The 5th edition was the first one with the built and stabilized questionnaire with a response rate over 90%. Therefore we have added that the results refer to the last edition both in the MS and in the caption of Figure 8.

- Your comment:
Conclusions: as the workshop is already in its 6th edition, some reflection could be done in terms of impact on the career development of former participants.

Response:
For the moment, though we do not have the structured and comprehensive data to respond to this pertinent suggestion, we do have ongoing collaborations with some former participants and we are aware that some have pursued research careers. Our intention is to prepare a survey and to gather feedback from the W.A.T.E.R. alumni after the completion of the 6th edition. That will help us to assess the short and long term impact of W.A.T.E.R. on the career development of the W.A.T.E.R. alumni.

- Your comment on typos:
Line 129: correct "instrumentssuch"  +  Line 143: correct "wherethe".

Response:
We have carefully checked the MS and corrected all the typos.

Sincerely,
Margaret Chen
On behalf of all co-authors